# Transcatheter Management of Pulmonary Sequestrations in Children—A Single-Center Experience

**DOI:** 10.3390/children10071197

**Published:** 2023-07-10

**Authors:** Ibrahim Abu Zahira, Raymond N. Haddad, Mathilde Meot, Damien Bonnet, Sophie Malekzadeh-Milani

**Affiliations:** 1M3C-Necker, Hôpital Universitaire Necker Enfants Malades, Assistance Publique-Hôpitaux de Paris (AP-HP), 75015 Paris, France; 2Faculté de Médecine, Université de Paris Cité, 15 Rue de l’École de Médecine, 75006 Paris, France

**Keywords:** congenital heart disease, children, endovascular embolization, pulmonary sequestration, transcatheter interventions

## Abstract

Background: A pulmonary sequestration (PS) is an area of bronchopulmonary tissue with aberrant arterial supply. Transcatheter occlusion of PSs is an appealing treatment option, but data on outcomes remain scarce. We aim to describe our experience with transcatheter management of PS in infants and children. Methods: Retrospective review of clinical data of all patients with suspected PS sent for diagnostic and/or interventional cardiac catheterization at our institution between January 1999 and May 2021. Procedural considerations, techniques, standard safety, and outcomes were assessed. Results: We identified 71 patients (52.1% males), with median age and weight of 4.9 months (IQR, 2.1–26.6) and 4.2 kg (IQR, 3.9–12.1), respectively. Sixty-one (86%) patients had associated congenital heart defects (CHDs). Forty-two (59%) patients had pulmonary arterial hypertension (PAH) at the time of diagnosis. Fifty-three (74.7%) patients underwent embolization of the PS feeding vessel using microcoils and/or vascular plugs, and eight (15.1%) of these were neonates who presented with severe PAH and cardiac failure. Two patients had large feeding vessels and were treated surgically. Sixteen (22.5%) patients with small feeding vessels received conservative management. At median follow-up of 36.4 months (IQR, 2.1–89.9), seven patients had died, 24 patients had CHD corrective surgeries, 26 patients had redo catheterizations, and five patients had persistent PAH. No PS surgical resection was needed, and no infection of the remaining lung tissue occurred. Conclusions: Transcatheter assessment and treatment of PSs is a safe and effective procedure. Neonates with large PSs are severely symptomatic and improve remarkably after PS closure. PS embolization and surgical repair of associated CHDs generally leads to the normalization of pulmonary pressures.

## 1. Introduction

Pulmonary sequestration (PS) is defined as non-functioning broncho-pulmonary tissue separated from the tracheobronchial tree and receiving blood supply from one or more abnormal systemic arteries. It is a rare congenital malformation that represents 0.15–6.4% of all pulmonary malformations [1,2,3,4,5]. PSs are classified as intra-lobar (ILS) or extra-lobar (ELS), based on the absence or presence of a distinct pleural covering [2,3,4]. Venous drainage is also frequently used to define the type of PS, as ILS cases almost always drain into the pulmonary veins (95%) whereas ELS PSs drain into the systemic veins in 75% of cases [2,6]. Bronchopulmonary sequestrations are increasingly diagnosed during pregnancy [7]. The main differential diagnosis is congenital airway pulmonary malformations, in which vascularization comes from pulmonary artery, and which presents as either multiple cysts, a single dominant cyst, or a solid mass with or without small multiple cysts. In utero, percutaneous laser therapy [8] or embolization may be proposed [9] for hydropic fetuses with congenital pulmonary lesions. Thoraco-amniotic shunt, tracheal decompression via laser perforation, and EXIT procedures are also possible rescue procedures [10]. In the post-natal period, ELS PSs often become symptomatic in the first months of life because of an important left-to-right shunt [2,11], whereas ILS cases are more frequently diagnosed in young adults with recurrent pneumonia [1,2]. Surgical resection of the abnormal tissue and ligation of its aberrant vascular supply used to be the conventional treatment for symptomatic PSs [6,12,13], with infrequent postoperative complications [12,13,14]. On the other hand, endovascular embolization of PSs has emerged as an effective treatment option and has been associated with reduced morbidity [12,15,16,17,18,19]. A hybrid approach combining endovascular treatment and surgical resection of abnormal tissue has been also described [12,20,21,22]. Current knowledge about embolization therapy remains limited to case reports and small case series [14,16,17,18,19,23,24,25]. The optimal treatment strategy for PSs in infants and children remains not clearly defined [5,6,17,26]. In this survey, the aim of the study was to report our institutional experience with transcatheter management of PSs in children and assess the short and mid-term efficacy and safety of this therapeutic option in the management of pulmonary sequestrations.

## 2. Materials and Methods

### 2.1. Patient Selection

The medical record of all patients undergoing diagnostic or interventional cardiac catheterization for suspected or known PS at our institution between January 1999 and May 2021 were retrospectively reviewed. PS was defined as a lung segment with no communication to the normal bronchial tree and vascularized by an aberrant systemic arterial supply. Patients were excluded if no PS was found. Neonates and younger patients, especially those who were highly symptomatic and with associated CHDs, were sent directly for cardiac catheterization for hemodynamic evaluation, confirmation of PS (suspected due to visualization of an abnormal vessel on cardiac ultrasound), and immediate endovascular embolization. Older paucisymptomatic patients often had chest CT angiography showing a PS. They were referred for elective cardiac catheterization to first confirm the diagnosis and occlude the feeding vessel when indicated.

### 2.2. Study Design

Procedural angiographies were retrospectively reviewed to collect the number of supplying vessels, PS location, and venous drainage of the malformation. We defined the type of PS according to venous drainage as ELS (PS draining into the systemic veins), ILS (PS draining into the pulmonary veins), and undetermined (PS with mixed drainage). Patient demographics, baseline, and follow-up clinical data were obtained from the patient records and were comprehensively analyzed. We then divided the children into two groups according to whether the sequestration had been embolized during first or second cath to determine any differences between these two populations.

### 2.3. Study Outcomes

We aim to describe the clinical characteristics of a population of children undergoing cardiac catheterization for pulmonary sequestration, to report the catheterization findings, efficacy, and safety, and to outline the short- and mid-term follow-up results.

### 2.4. Catheterization Procedure

All procedures were performed in the catheterization laboratory under general anesthesia, systemic heparinization, and biplane fluoroscopy. Patients received antibiotic prophylaxis when a device was implanted. After hemodynamic evaluation and angiography of the pulmonary arteries, an aortogram was performed to map the systemic aberrant arteries. Selective cannulation of each aberrant artery was performed using standard diagnostic catheters in large vessels or microcatheters and coronary wires in smaller ones. Selective hand-angiography of abnormal vessels was performed to determine the size and location of the PS, analyze the venous drainage, and choose the appropriate device for closure (microcoils, vascular plugs, or a combination of both). Device selection changed throughout the study period according to the availability of new generation low-profile vascular plugs, such as the Amplatzer Vascular Plug (AVP) (Abbott, Chicago, IL, USA) and the later Microvascular Plug (MVP) (Medtronic, Minneapolis, MN, USA) [27]. In some cases, low-profile vascular plugs were intentionally combined with microcoils to ensure adequate vessel closure. This approach was more commonly used in small babies to target larger vessels using smaller devices and thereby avoiding larger arterial access. This approach was also applied in extremely tortuous vessels where microcatheters were needed to achieve super-selective cannulation, thereby avoiding the instability of larger delivery systems encountered with bulkier devices. All embolizations were controlled by selective exit angiographies to confirm closure. The implantation was considered successful only when the occluder device was released into the desired position without immediate complication. Two interventional cardiologists, with 25 and 15 years’ experience of pediatric cardiac catheterization, were in post during the inclusion period.

### 2.5. Statistical Analysis

Statistical analyses were performed using Statistical Package for the Social Sciences Statistics (SPSS) version 22 for Macintosh (IBM, Armonk, NY, USA). Categorical variables are reported as frequency and percentage. Continuous variables are presented as median with interquartile range (IQR). The normality of measurements was assessed using the Shapiro–Wilk test. Categorical variables were compared using a chi-square test or Fisher’s exact test, and continuous variables using the Mann–Whitney U test. Tests were considered significant for a *p*-value < 0.05.

## 3. Results

### 3.1. Patients

We identified 71 patients (52.1% males) with a median procedural age of 4.9 months (IQR, 2.1–26.6) and median weight of 4.2 kg (IQR, 3.9–12.1). Of these, 10 (14.1%) patients were neonates and 41 (57.7%) were infants. The patients’ characteristics are outlined in Table 1. Of the 71, 61 (85.9%) patients had associated congenital heart defects (CHDs), of which 47 (77%) patients had scimitar syndrome. In total, 49 (69%) patients had pulmonary arterial hypertension (PAH) at time of catheterization. Four patients had stenotic scimitar veins, and four patients had stenosis of at least one of the pulmonary veins normally connecting to the left atrium. Associated CHDs are shown in Appendix A. Extracardiac anomalies were found in 13 (18.3%) patients, with four patients having a right congenital diaphragmatic hernia.

### 3.2. Catheterization Procedure

Catheterization data are presented in Table 2. Pulmonary artery branches were normal in 34 (48%) patients while 32 (45.1%) patients had hypoplasia of the right pulmonary artery, mostly in relation to scimitar syndrome. A total of 93 abnormal supplying vessels were identified in 71 patients, with 16 (22.5%) patients having two or more aberrant systemic arteries. The abnormal arterial supply originated from the abdominal aorta in 90.3% of cases. The PS was located on the right side in 81.7% of cases. The PSs were ILS in 48.4% and ELS in 46.2% of cases. Five (5.4%) PSs were undetermined in type. Patients with CHDs had various types of PS, while patients without CHDs had ILS PS exclusively.

### 3.3. Endovascular Embolization

During the first procedure, 47/71 (66.2%) patients underwent endovascular embolization of the PS feeding vessel (Figure 1, Figure 2 and Figure 3). The type and number of occluder vascular devices are detailed in Table 3. In 14/47 (36.2%) patients, occlusion was achieved with a combination of vascular plugs and micro coils (Figure 1, Figure 2 and Figure 3). Reasons for non-embolization or delayed embolization are also described in Table 3. In 21 cases, the feeding vessels were too small or without hemodynamic significance. The feeding vessels were too large for endovascular treatment (no suitable device available on the market at the time of catheterization) in two patients (Figure 4), and one patient was hemodynamically unstable, leading to abortion of the embolization procedure. One coil migrated and was recaptured by snare. One patient experienced transient atrioventricular block during right heart catheterization. Procedure time and radiation exposure are detailed in Table 3. The median hospital stay was 3 days (IQR, 2–8).

### 3.4. Follow-Up and Outcomes

At median follow-up of 36.4 months (IQR, 2–90), seven patients (9.8%) had died. One 45-day-old infant with severe PAH died from a pulmonary hypertensive crisis during a redo procedure when we attempted to dilate a stent previously placed within a stenotic scimitar vein. Four patients died from heart failure, severe PAH, and severe respiratory failure before 6 months of age. One patient died at 3.8 years of age after heart transplantation for hypoplastic left heart syndrome. One patient with Eisenmenger physiology died from massive hemoptysis at 18 years of age. Among the survivors, 5/64 (7.8%) patients had persistent PAH at last follow-up, of which three are receiving PAH therapies. Most patients with chronic or recurrent chest infections became asymptomatic after PS embolization. No patient had surgical removal of the abnormal lung tissue, and no PS-related infection was documented. The two large feeding vessels were ligated surgically because embolization was not technically feasible (no suitable device available at that time or the device required a delivery system too large for the child’s weight). Of the other patients, 24 (33.8%) had corrective surgeries for associated CHDs within a median period of 3.2 months (IQR, 1–11) after cardiac catheterization, and 26 (36.6%) patients had redo cardiac catheterizations, of which 14 (53.9%) patients had successful repeated PS embolizations. The remaining 12 patients underwent redo cardiac catheterizations for various diagnostic purposes. The reasons for repeated endovascular embolization of the PSs were recanalization (n = 6), residual shunt (n = 2), and previously non-embolized vessels (n = 6). Overall, 16 (22.5%) patients with small feeding vessels received conservative management of the PSs. At median follow-up of 48 months (IQR, 16–118), 7/16 patients had corrective surgeries for associated CHDs, and all are alive with good outcomes. Results and follow-up are shown in Figure 5.

## 4. Discussion

Putting our study findings into perspective, we observed a wide spectrum of phenotypic PS presentation, ranging from severely symptomatic neonates with cardiac overload and PAH to paucisymptomatic older patients [5,17]. The population can be divided into two sub-groups. In neonates, closure of the feeding vessel often leads to spectacular improvement related to the suppression of the left-to-right shunt and its role in the elevation of pulmonary artery pressures. In cases of associated CHDs, suppression of the shunt may facilitate the subsequent management of the CHD by delaying surgical repair and thereby improving the overall outcomes of the patient [13,28,29]. The second subgroup includes infants with left-to-right shunts, paucisymptomatic patients, and patients with recurrent chest infections. In this subgroup, elective shunt closure solves the problem most of the time, and outpatient follow-up is needed to exclude residual shunt or recurrent infections. In this cohort, 16 (22.5%) patients with small feeding vessels had conservative management of the PSs and no embolization was performed. The evolution was unremarkable with good medium-term follow-up. This supports the idea that small and hemodynamically insignificant PSs can be neglected. These patients were older than those who were actively treated and often had non-specific symptoms. We did not observe, unlike previously reported, hemoptysis as a presenting symptom or during follow-up [30,31]. In these patients, diagnosis can be difficult without radiological investigation for vasculature and pulmonary parenchyma.

We found that ELS PSs were more frequent than in previous series [2,3,4], with only 48.4% of cases being ILS, probably because of the high proportion of patients with associated CHDs. However, it is noteworthy that the classification we adopted to determine ILS or ELS based on PV drainage is not fully accurate. This classification is usually used by clinicians as even chest CT scans often fail to show the pleural covering. As in the series by Brown et al. [17], the majority of PSs were right sided in our cohort, which is unquestionably related to the predominance of scimitar syndrome. The aberrant feeding arteries originated from the abdominal aorta in 90.3% of patients, also differing from previous cohorts in which arterial supply vessels more frequently originated from the thoracic aorta [2,3].

Endovascular embolization of PSs causes necrosis, followed by progressive fibrosis and involution of the abnormal lung tissue. Moreover, the retained non-aerated pulmonary parenchyma can become a nidus for infection and abscess formation, but this was not shown in this series. Our data shows that the transcatheter approach is an effective, minimally invasive treatment option for symptomatic PS [17]. Management of asymptomatic PS is still debated [13]. Persistent shunt closure was observed in all treated patients (with the need for redo catheterization in some), and no complications related to the lung tissue left in place occurred during follow-up.

Shunt persistence has been reported after treatment with endovascular embolization [25] due to multiple feeding vessels, incomplete closure, displacement of the embolic agents, or opening of collateral circles. Therefore, routine outpatient follow-up is essential for all patients. Chest CT angiograms or repeated cardiac catheterization is only necessary in the presence of hemodynamically significant lesions.

Microcoils (Figure 1), vascular plugs (Figure 2), and a combination of both (Figure 3) were our devices of choice to embolize the PS-feeding arteries. Polyvinyl alcohol (PVA) particles, gelatin sponge, alcohol, glue (n-butylcyanacrylate), and various combinations of embolic materials have also been reported [12,16,17,25,27]. However, the high-flow nature of such anomalies and their drainage into pulmonary veins in many patients require embolic products with a low likelihood of distal embolization or migration. Microcoils are currently the most used embolic agents for vessel anomalies in many institutions. However, super-selective endovascular occlusions using microcoils can sometimes be challenging due to anatomic considerations. In high-flow arterial vessels such as PSs, numerous coils can be required for complete occlusion. On the other hand, new-generation low-profile microvascular plugs allow us to treat these vessels using microcatheters with diameters of 6–7 mm [27]. Using these devices, we were able to track smoothly through tortuous vessels and keep arterial access at a maximum of 5 Fr. A combination of microvascular plugs and microcoils has been also effective in larger feeding vessels, where we packed the vascular plugs with long microcoils if persistent flow was demonstrated by control angiographies.

The overall mortality rate was relatively high in our population, and this was mainly related to multifactorial PAH and/or an association with severe CHD [32]. In general, the common evolution of PAH is favorable after PS embolization. In our series, 26 (72.2%) of the 36 severely ill patients with confirmed PAH before PS embolization demonstrated complete normalization or significant decrease in disease severity at last follow-up. In neonates with suprasystemic hypertension at time of diagnosis, the prognosis is poor despite PS embolization. This is mainly due to associated anomalies such as pulmonary venous return obstruction.

Optimal management of symptomatic and asymptomatic PS remains to be defined. Open surgical resection is the current standard and curative treatment for PS even in asymptomatic cases, but embolization has had a significant place for 20 years in our institution as an alternative to or in conjunction with surgery [5,6,13]. One of the advocated reasons to perform surgical resection instead of embolization is the development of malignancy or PAH in abnormal lung tissue, but this is debatable and no robust data support this idea in PS (unlike in cystic pulmonary airway malformation). The alternative surgical approach is minimally invasive video-assisted lobectomy, but this can be quite challenging in very large feeding arteries.

There may be some overlap between the study by Khen Dunlop et al. and our own insofar as the patients were treated at the same center. Nevertheless, as most of our patients had associated congenital heart disease, they were often referred directly to the cardiology department and not the thoracic surgery department. Patients with isolated pulmonary sequestration (without congenital heart disease) could have been included in both studies as some of the patients included in Khen Dunlop et al. may have been referred to the interventional radiologist for embolization until 2015.

In this study, we have shown that endovascular suppression of PS is a feasible, safe, and efficient therapy in young and older patients with no procedure-related morbidity or mortality and no or few retained non-aerated lung tissue-related complications. No patients in this study required surgery after embolization.

### Limitations

This was a single-center retrospective study covering 22 years of experience. Our practice has varied during the study period, depending on our knowledge regarding PSs and the occluding devices in the armamentarium. The data were collected from our cardiac catheterization database, which probably creates a selection bias of patients and explains the high prevalence of CHDs in our cohort and the proportion of newborns with hemodynamic compromise. This proportion might change and be more comparable to those in previous series through expanded collaboration with other pediatric pulmonologists. Reported symptoms may not necessarily be related to PS but more to the associated CHDs. It may be argued that PS diagnosis can be made by chest CT angiography, especially in older patients, as this approach can best demonstrate parenchymal abnormalities along with aberrant vascular supply. However, we believe that a conventional angiogram remains an interesting first-step diagnostic tool, especially if embolization is planned in small children. In older patients, CT remains an important diagnostic tool.

## 5. Conclusions

In this single-center series, we found that transcatheter assessment and treatment of PSs is a safe and effective procedure with good outcomes. PS is frequently associated with complex CHDs, and prognosis depends on both the presence of PAH and the severity of the CHD. Early closure of PS can be lifesaving in neonates who usually present with cardiac failure and/or severe PAH. In older patients, endovascular treatment without surgical resection of abnormal lung tissue is safe and effective in our experience, with satisfactory follow-up. Larger series and meta-analyses could help define optimal management for pulmonary sequestrations.

## Figures and Tables

**Figure 1 children-10-01197-f001:**
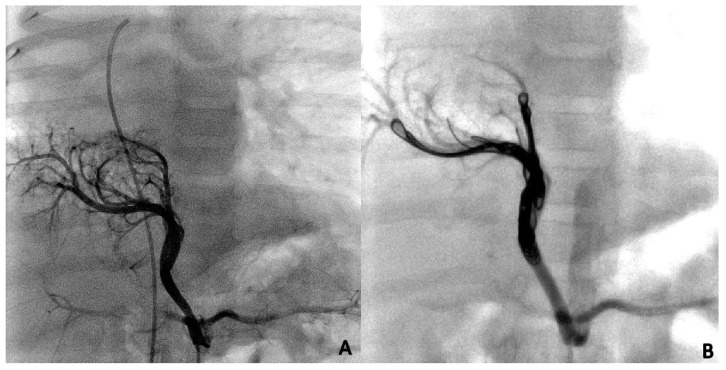
Selective angiographies before (**A**) and after (**B**) closure of the PS feeding vessel using multiple Concerto microcoils (Medtronic, Minneapolis, MN, USA) of various sizes.

**Figure 2 children-10-01197-f002:**
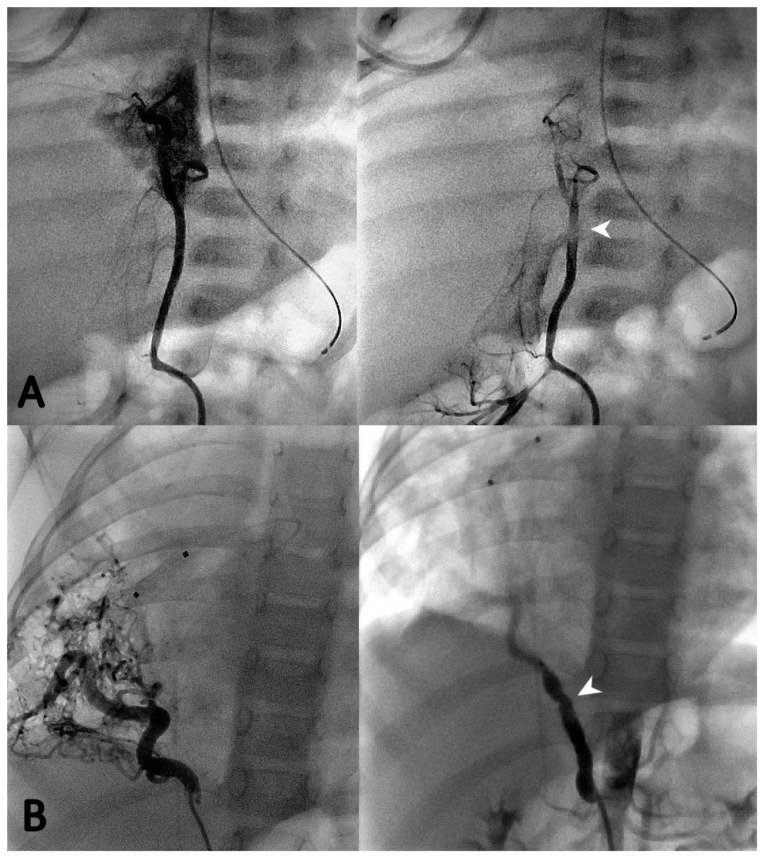
Two cases of selective angiography before (**left** panel) and after (**right** panel) closure of the PS feeding vessel using one Microvascular Plug-7Q (Medtronic, Minneapolis, MN, USA) (**A**) and one 7 mm Amplatzer Vascular Plug IV (St Jude Medical^®^, Minneapolis, MN, USA) (**B**) (white arrows).

**Figure 3 children-10-01197-f003:**
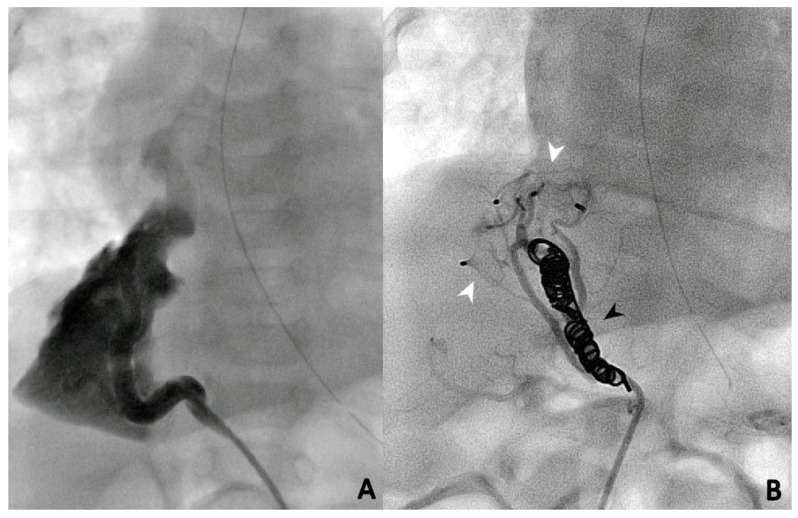
Selective angiographies before (**A**) and after (**B**) closure of the PS feeding vessel using a combination of two Amplatzer vascular plugs (St Jude Medical^®^, Minneapolis, MN, USA) (white arrows) and two Concerto microcoils (Medtronic, Minneapolis, MN, USA) (black arrows).

**Figure 4 children-10-01197-f004:**
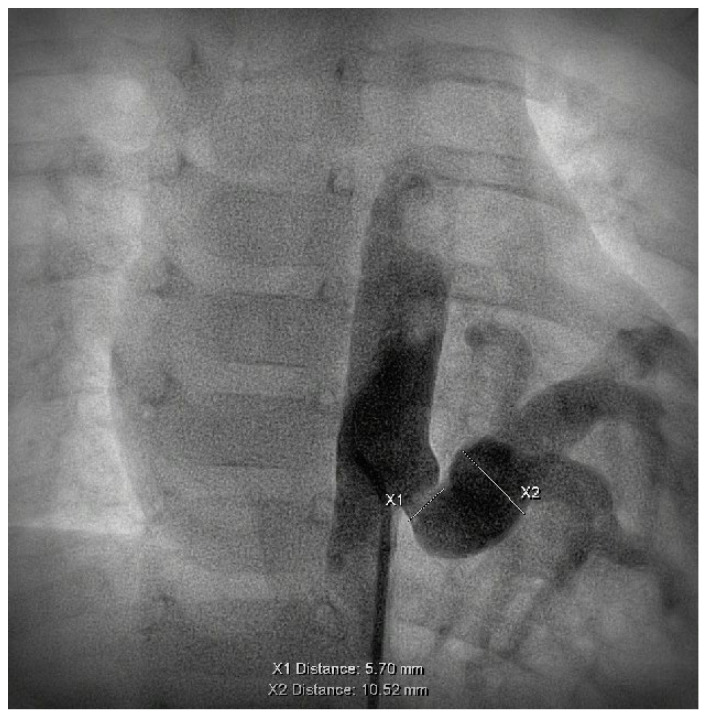
Large vessel, not suitable for endovascular treatment, at time of procedure.

**Figure 5 children-10-01197-f005:**
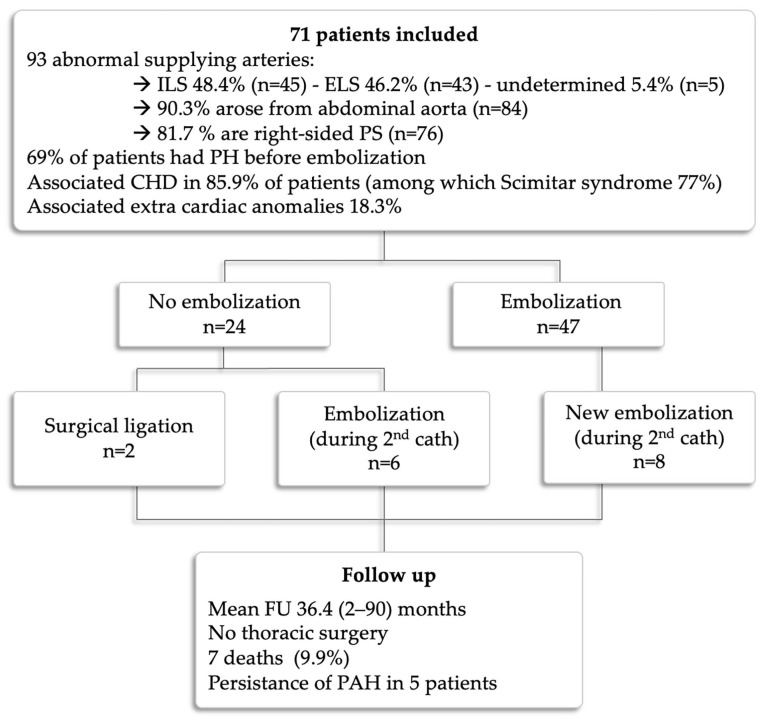
Patient flow chart.

**Table 1 children-10-01197-t001:** Patients’ baseline clinical characteristics.

	Total	Non-Embolized	Embolized ^†^	*p*
	N = 71	N = 18	N = 53	
Male, N (%)	37 (52.1)	10 (55.6)	27 (50.9)	0.79 ^a^
Age (months), median (IQR)	4.9 (2.1–26.6)	16.1 (2.9–50.5)	4.3 (1.9–11.6)	0.07 ^b^
Age groups, N (%)				0.928 ^c^
Neonates (0–30 days)	10 (14.1)	2 (11.1)	8 (15.1)	
Infants (1–12 months)	33 (46.5)	8 (24.3)	25 (75.7)	
Young children (1–6 years)	17 (23.9)	5 (29.4)	12 (70.6)	
Children (6–12 years)	6 (8.5)	2 (11.1)	4 (7.5)	
Adolescents (12–18 years)	5 (7)	1 (5.6)	4 (7.5)	
Weight (kg), median (IQR)	4.2 (3.9–12.1)	9 (4.9–19)	4.2 (3.2–8.8)	**0.007 ^c^**
Body surface area (m^2^), median (IQR)	0.25 (0.24–0.54)	0.43 (0.28–0.76)	0.25 (0.21–0.43)	**0.007 ^c^**
Prenatal diagnosis of pulmonary sequestration, N (%)	7 (9.9)	1 (5.6)	6 (11.3)	0.67 ^c^
Prenatal diagnosis of associated heart defects, N (%)	13 (18.3)	4 (22.2)	9 (17)	0.726 ^c^
Associated congenital heart diseases, N (%)	61 (85.9)	15 (83.3)	46 (86.8)	0.706 ^c^
Associated non-cardiac anomalies/malformations, N (%)	13 (18.3)	6 (33.3)	7 (13.2)	0.079 ^c^
Heart position on chest X-ray, N (%)				0.749 ^c^
Levocardia	30 (42.2)	8 (44.4)	22 (41.5)	
Dextrocardia	32 (40.1)	7 (38.9)	25 (47.2)	
Mesocardia	9 (12.7)	3 (16.7)	6 (11.3)	
Clinical presentation/symptoms *, N (%)				
Asymptomatic/incidental diagnosis/persistent dry cough	13 (18.3)	5 (27.8)	8 (15.1)	0.292 ^c^
Cyanosis	12 (16.9)	1 (5.6)	11 (20.8)	0.273 ^c^
Heart murmur	32 (45.1)	6 (33.3)	26 (49.1)	0.284 ^a^
Failure to thrive	16 (22.5)	2 (11.1)	14 (26.4)	0.327 ^c^
Mild-to-moderate respiratory distress	21 (26.9)	1 (5.6)	20 (37.7)	**0.014 ^a^**
Chronic or recurrent chest infection	27 (38)	5 (27.8)	22 (41.5)	0.403 ^a^
Mild-to-moderate symptoms of heart failure	11 (15.5)	1 (5.6)	10 (18.9)	0.269 ^c^
Pulmonary hypertension	42 (59.1)	8 (44.4)	34 (64.2)	0.171 ^a^
Respiratory failure	9 (12.7)	1 (5.6)	8 (15.1)	0.432 ^c^
Intubation, N (%)	9 (12.7)	1 (5.6)	8 (15.1)	0.432 ^c^
Nitrite oxide therapy, N (%)	2 (2.8)	--	2 (3.8)	--
Baseline ultrasound findings *, N (%)				
Normal pulmonary artery pressure	29 (40.8)	10 (55.6)	19 (35.8)	
Pulmonary hypertension	42 (59.2)	8 (44.4)	34 (64.2)	
Infra-systemic	18 (25.4)	7 (38.9)	11 (20.8)	
Iso-systemic	9 (12.7)	--	9 (17)	**0.025 ^c^**
Supra-systemic	15 (21.1)	1 (5.6)	14 (26.4)	
Dilated right ventricle	39 (54.9)	7 (38.9)	32 (60.4)	0.17 ^a^
Right ventricular dysfunction	6 (8.5)	1 (5.6)	5 (9.4)	1 ^c^
Dilated left ventricle	10 (14.1)	1 (5.6)	9 (17)	0.434 ^a^

IQR = interquartile range; ^a^*:* chi-square test; ^b^: Mann–Whitney U test; ^c^: Fisher’s exact test. Bold values are significant *p*-values; ^†^ during study period; * multiple choice applied.

**Table 2 children-10-01197-t002:** Angiographic and hemodynamical data.

**Pulmonary artery anatomy/anomaly, N (%), n = 71**	
Normal anatomy	34 (47.9)
RPA agenesis	2 (2.8)
RPA hypoplasia	32 (45.1)
LPA hypoplasia	3 (4.2)
**Number of individual supply arteries per patient, N (%), n = 71**	
1	55 (77.5)
2	13 (18.3)
4	3 (4.2)
**Arterial blood supply origin, N (%), n = 93**	
Abdominal aorta	84 (90.3)
Thoracic aorta	9 (9.7)
**Side of pulmonary sequestration, N (%), n = 93**	
Left lung	9 (9.7)
Right lung	76 (81.7)
Both lungs	8 (8.6)
**Distribution of pulmonary sequestration, N (%), n = 93**	
Lower left lobe	14 (15.1)
Right lower lobe	71 (76.3)
Right upper lobe	3 (3.2)
Right middle lobe	5 (5.4)
**Type of pulmonary sequestration, N (%), n = 93**	
Intra-lobar	45 (48.4)
Extra-lobar	43 (46.2)
Mixed	5 (5.4)
**Associated anomalous pulmonary vein into the systemic venous system, N (%), n = 71**	47 (66.2)
Non-obstructed	37 (78.7)
Occluded	4 (8.5)
Stenotic	4 (8.5)
Atypical	2 (4.3)
**Pulmonary artery pressure, N (%), n = 71**	
Normal	22 (31)
Pulmonary hypertension	49 (69)
Infra-systemic	29
Iso-systemic	9
Supra-systemic	11

**Table 3 children-10-01197-t003:** Procedural related data for the first catheterization.

Planned embolization post-corrective surgery, N (%), n = 71	5 (7)
Embolization, N (%), n = 71	47 (66.2)
Vascular plugs/occluder devices	14 (29.8)
Coils/microcoils	16 (34)
Combination of embolization material	17 (36.2)
Number of vascular plugs/occluder devices per setting, median (IQR), n = 47	1 (1–2)
Number of coils/microcoils per setting, median (IQR), n = 47	2 (1–3)
Number of embolized supply vessels during the same procedure, N (%), n = 47	
1	33 (70.2)
2	11 (23.4)
3	3 (6.4)
Reason for non-embolization or delayed embolization *, N (%), n = 24	
Small supply vessel/hemodynamical insignificance	16 (66.7)
Supply vessel too large	2 (8.3)
Underestimation/re-evaluation of hemodynamical significance	5 (20.8)
Patient instability/anesthesia problems	1 (4.2)
Re-embolization during second/third catheterization, N (%), n = 71	14 (19.7)
Previously non/delayed embolized vessel supply, N (%), n = 24	6 (25)
Previously embolized vessel supply (residual shunt), N (%), n = 47	2 (4.3)
Previously embolized site (recanalization), N (%), n = 47	6 (12.8)
Procedure time (min) ^‡^, median (IQR), n = 57 ^⫲^	60 (40–90)
Fluoroscopy time (min) ^‡^, median (IQR), n = 61 ^⫲^	16.9 (9.5–24.8)
Total dose area product (µGy.m^2^) ^‡^, median (IQR), n = 50 ^⫲^	264.8 (147.5–663.7)
K_ar_ (mGy) ^‡^, median (IQR), n = 58 ^⫲^	64 (31.5–127.2)

IQR = interquartile range; * multiple choice applied; ^‡^ related to first cardiac catheterization; ^⫲^ reduced effective related to missing data; K_ar_: cumulative air kerma at patient entrance reference point.

## Data Availability

The raw data supporting the conclusions of this article will be made available by the authors, upon request, to any qualified researcher.

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
