# Peer review of "Transcatheter Management of Pulmonary Sequestrations in Children—A Single-Center Experience"

_children, 2023, doi:10.3390/children10071197_

Round 1
Reviewer 1 Report
Good paper, large patient cohort with pulmonary sequestration and probably the only large series of scimitar syndrome and bronchopulmonary sequestration. Written from the perspective of pediatric cardiologist.
Part of the cohort probably included in the series (46 BP sequestrations receiving embolization, 2000-2015) reported by Khen-Dunlop et al. (Eur J Cardio-Thoracic Surg 54:246-251, 2018), giving the pediatric surgeon`s point of view from the same institution.
Authors should comment on possible overlap and if there is any, they should refer to this study (which is neither mentioned nor in the reference list).
Author Response
Point 1: Good paper, large patient cohort with pulmonary sequestration and probably the only large series of scimitar syndrome and bronchopulmonary sequestration. Written from the perspective of pediatric cardiologist.
Part of the cohort probably included in the series (46 BP sequestrations receiving embolization, 2000-2015) reported by Khen-Dunlop et al. (Eur J Cardio-Thoracic Surg 54:246-251, 2018), giving the pediatric surgeon`s point of view from the same institution. Authors should comment on possible overlap and if there is any, they should refer to this study (which is neither mentioned nor in the reference list).
Response 1: Thank you for your comment. There is indeed a potential overlap between the two studies. However, the current cohort is composed mainly of patients with associated congenital heart disease, whereas the Khen Dunlop et al. cohort was composed mainly of isolated pulmonary sequestrations.
We have added the reference and mentioned the overlap in the discussion section.
Reviewer 2 Report
I suggest the following title: "Transcatheter Management of Pulmonary Sequestrations in Children - a Single-Center Experience"
In the introduction, you must also touch on fetal surgery (fetal thoracoamniotic shunt placement, laser coagulation, EXIT procedure, etc.) so that readers have insights into all treatment options.
Also, in two or three sentences in the introduction, touch on the differential diagnostic conditions.
At the end of the introduction, define more clearly the main and secondary objectives of your study.
I suggest that instead of the section "Patient selection and study design" you form the following sections; "Patient selection", "Study design", and "Study outcomes".
Remove the following sentences "All procedures contributing to this …. 1975, as revised in 2008. Approval from the Institutional …. (MR004: 2022- 0311142602). Informed consent was …. inclusion in the study." considering that the mentioned information is at the end of the manuscript.
When describing the catheterization procedure for commonly known information, reduce the textual content by adding references that describe the procedure. You must also clearly state how many operators participated in the treatment during the specified period.
The "Statistical Analysis" section is insufficient. You must clearly state which tests were used to obtain the results.
Be accurate and careful in displaying the results. Are infants from 1 to 24 months old?
In the "Materials and Methods" section, describe more clearly how you formed the groups.
I definitely suggest that you include one or more flowcharts as part of the results to make it easier to follow patients.
I suggest you put table 2 in "supplementary materials".
Please recheck all results as there are a lot of incorrect data (for example....9.7 + 94.9 + 8.6 = 113.2%,...etc...)
You must clearly specify what is a large vessel that is not an indication for embolization.
Given that these are the results of one center, which results in a smaller sample, be careful in drawing conclusions. Form the conclusion with an emphasis on your own observations and results, and not so that the readers get the impression that this is the most optimal method. Accurate and specific conclusions will be drawn from systematic reviews and meta-analyses.
Moderate English language editing required.
Author Response
Point 1: I suggest the following title: "Transcatheter Management of Pulmonary Sequestrations in Children - a Single-Center Experience"
Response 1: Thank you for your suggestion, we made the change.
Point 2: In the introduction, you must also touch on fetal surgery (fetal thoracoamniotic shunt placement, laser coagulation, EXIT procedure, etc.) so that readers have insights into all treatment options.
Response 2: We added the following sentences in the introduction : “In utero, percutaneous laser therapy or embolization may be proposed for hydropic fetuses with congenital pulmonary lesions. Thoraco-amniotic shunt, tracheal decompression via laser perforation and EXIT procedure are also possible in utero rescue procedures.”
Point 3: Also, in two or three sentences in the introduction, touch on the differential diagnostic conditions.
Response 3: We also added the following sentences : “Bronchopulmonary sequestrations are increasingly diagnosed during pregnancy. The main differential diagnosis is congenital airway pulmonary malformations, whom vascularization arises from pulmonary arteries, and which presents as either multiple cysts, a single dominant cyst or a solid mass with or without small multiple cysts.”
Point 4: At the end of the introduction, define more clearly the main and secondary objectives of your study.
Response 4: As suggested, we defined more clearly the main and secondary objectives of your study as follows : “We thought to report our institutional experience with transcatheter management of PS in small patients and assess the short and mid-term efficacy and safety of this therapeutic option in the management of PS”.
Point 5: I suggest that instead of the section "Patient selection and study design" you form the following sections; "Patient selection", "Study design", and "Study outcomes".
Response 5: We made the change.
Point 6: Remove the following sentences "All procedures contributing to this …. 1975, as revised in 2008. Approval from the Institutional …. (MR004: 2022- 0311142602). Informed consent was …. inclusion in the study." considering that the mentioned information is at the end of the manuscript.
Response 6: We removed these sentences, as suggested.
Point 7: When describing the catheterization procedure for commonly known information, reduce the textual content by adding references that describe the procedure.
Response 7: We added the following reference : Haddad RN, Bonnet D, Malekzadeh-Milani S. Embolization of vascular abnormalities in children with congenital heart diseases using medtronic micro vascular plugs. Heart Vessels. 2022 Jul;37(7):1271-1282. doi: 10.1007/s00380-021-02007-6.
Nevertheless, we have not reduced the text too much, since many articles on vessel embolization concern adult patients, and we felt it was important to emphasize the technical aspects specific to pediatrics.
Point 8: You must also clearly state how many operators participated in the treatment during the specified period.
Response 8: We added this information in the method section as follows : “Two interventional cardiologists, with 25- and 15-years’ experience of pediatric cardiac catheterization, took over during the inclusion period”
Point 9: The "Statistical Analysis" section is insufficient. You must clearly state which tests were used to obtain the results.
Response 9 : We changed the section as follows: “categorical variables were compared using chi-square test, of Fisher exact test, and continuous variables using Mann-Whitney U test. Tests were considered significant for a p-value < 0.05”
Point 10: Be accurate and careful in displaying the results. Are infants from 1 to 24 months old?
Response 10: We apologize for the mistake, we changed the table 1.
Point 11: In the "Materials and Methods" section, describe more clearly how you formed the groups.
Response 11: For better clarity we added the following description of the group : “We then divided the children into two groups according to whether or not the sequestration had been embolized, to see if there were any differences between these two populations”.
Point 12: I definitely suggest that you include one or more flowcharts as part of the results to make it easier to follow patients.
Response 12: We added a flow chart as figure 5.
Point 13: I suggest you put table 2 in "supplementary materials".
Response 13: We made the change.
Point 14: Please recheck all results as there are a lot of incorrect data (for example....9.7 + 94.9 + 8.6 = 113.2%,...etc...)
Response 14: We apologize for the mistake and rechecked all the results.
Point 15: You must clearly specify what is a large vessel that is not an indication for embolization.
Response 15: For better clarity, we added the following sentence in the Results section : “The two large feeding vessels were ligated surgically because embolization was not technically possible (no suitable device available or device requiring a delivery system too large for the child's weight).”
Point 16: Given that these are the results of one center, which results in a smaller sample, be careful in drawing conclusions. Form the conclusion with an emphasis on your own observations and results, and not so that the readers get the impression that this is the most optimal method. Accurate and specific conclusions will be drawn from systematic reviews and meta-analyses.
Response 16: We change the conclusion as follow : “In this single-center series, transcatheter assessment and treatment of PSs is a safe and effective procedure with good outcome. […] Larger series and meta-analyses could help define optimal management of pulmonary sequestrations.”

Round 2
Reviewer 2 Report
You do not need to separate the sections in the abstract. Pay attention to keywords (no numbers).
The introduction is now sufficient, but pay attention to the order of the references that do not follow one another (eg after reference 11 you cite reference 13).
In defining aims, it is awkward to use the expression "...we thought...". Be clearer. Rather start the sentence..."The aim of the study was...".
Describe the "Study outcomes" section in more detail in relation to the results you presented.
There are values in the newly created flowchart that do not match the values in the text. Please correct and check everything again!
Table 2 is now Table S1, and Table 3 is now Table 2.
Please double-check all the data in the tables. For example, sum of percentages (15+76.3+3.2+5.4=99.9). Please be accurate!
Moderate editing of English language required
